# Error Model for the Assimilation of All-Sky FY-4A/AGRI Infrared Radiance Observations

**DOI:** 10.3390/s24082572

**Published:** 2024-04-17

**Authors:** Dongchuan Pu, Yali Wu

**Affiliations:** 1School of Environment, Harbin Institute of Technology, Harbin 150006, China; pudc2020@mail.sustech.edu.cn; 2School of Environmental Science and Engineering, Southern University of Science and Technology, Shenzhen 518055, China; 3Guangdong-Hong Kong-Macao Greater Bay Area Weather Research Center for Monitoring Warning and Forecasting (Shenzhen Institute of Meteorological Innovation), Shenzhen 518016, China

**Keywords:** observation error, cloud effect, assimilation, FY-4A/AGRI, infrared radiance

## Abstract

The Advanced Geostationary Radiation Imager (AGRI) carried by the FengYun-4A (FY-4A) satellite enables the continuous observation of local weather. However, FY-4A/AGRI infrared satellite observations are strongly influenced by clouds, which complicates their use in all-sky data assimilation. The presence of clouds leads to increased uncertainty, and the observation-minus-background (*O*−*B*) differences can significantly deviate from the Gaussian distribution assumed in the variational data assimilation theory. In this study, we introduce two cloud-affected (*Ca*) indices to quantify the impact of cloud amount and establish dynamic observation error models to address biases between *O*−*B* and Gaussian distributions when assimilating all-sky data from FY-4A/AGRI observations. For each *Ca* index, we evaluate two dynamic observation error models: a two-segment and a three-segment linear model. Our findings indicate that the three-segment linear model we propose better conforms to the statistical characteristics of FY-4A/AGRI observations and improves the Gaussianity of the *O*−*B* probability density function. Dynamic observation error models developed in this study are capable of handling cloud-free or cloud-affected FY-4A/AGRI observations in a uniform manner without cloud detection.

## 1. Introduction

Earth geostationary satellite observation data play a crucial role in regional-scale numerical weather forecasting [1]. China’s geostationary satellite Fengyun-4A (FY-4A), equipped with the Advanced Geostationary Radiation Imager (AGRI) [2], provides high spatiotemporal resolution atmospheric information [3]. In the infrared (IR) band of FY-4A/AGRI, radiation absorption and emission are highly sensitive to atmospheric temperature, humidity, and cloud precipitation [4]. Nonlinear impacts of clouds on FY-4A/AGRI observations, along with systematic biases in assimilation models, pose significant challenges for all-sky data assimilation techniques [5]. These issues result in non-Gaussian distributions of the observation-minus-background (*O*−*B*) differences and violate the fundamental assumptions of mainstream variational data assimilation methods [6]. Therefore, it is essential to address observational errors in FY-4A/AGRI observations and enhance the robustness and analysis quality of all-sky data assimilation systems.

The presence of clouds often profoundly affects the quality and assimilation effectiveness of satellite observation [7]. Considering the significant challenges in assimilating cloud-affected data, the most common approach currently is to preprocess satellite observation data, removing data influenced by cloud layers [1,8,9]. However, cloud-affected satellite radiance data, unrestricted by space–time constraints, can rapidly capture the occurrence, development, and dissipation of convection, making it one of the most valuable sources of observational data for regional numerical forecasting, potentially significantly improving the accuracy of high-resolution numerical forecasts [10]. Therefore, many studies continue to attempt various methods to assimilate this cloudy radiance data [11].

Currently, there are two main approaches to assimilate cloudy radiance data [9]. The first approach involves using cloud detection algorithms to distinguish between clear-sky and cloudy areas and then assimilating clear-sky and cloudy data separately, with the latter typically including satellite-retrieved cloud products [11]. However, due to the high uncertainty of retrieval algorithms, the statistical characteristics of satellite cloud product observation errors are complex [12]. Another approach is to directly assimilate satellite observation data with and without clouds within a unified framework, without the need for cloud detection [13].

Cloud effect observation error models have been developed to improve the effectiveness of all-sky data assimilation [8,14]. Geer et al. [15] proposed an observation error model based on cloud fraction. Since the forward observation operator error of the assimilation system is significant under cloudy conditions, the observation error tends to increase with larger cloud fractions. By utilizing this basic concept and employing a cloud effect dynamic observation error estimation method, an approximately Gaussian *O*−*B* distribution can be obtained, simultaneously enhancing quality control and greatly improving the effectiveness of all-sky satellite data assimilation. Particularly in the assimilation of infrared satellite data, the application of cloud effect observation error models can enhance the understanding of atmospheric states [16], leading to higher-precision short-term forecast results [17]. Okamoto et al. [18] proposed cloud-affected (*Ca*) indices and a two-segment linear observation error model for assimilating Advanced Himawari Imager (AHI) radiance data under cloudy conditions. The probability density function of the infrared band *O*−*B* exhibits a near-Gaussian distribution, improving the accuracy of rainfall forecasts [16,19]. Harnisch et al. [20] defined a new cloud effect parameter model using a threshold method for assimilating Spinning Enhanced Visible and InfraRed Imager (SEVIRI) infrared observation data, obtaining more robust estimates of cloud impacts. Geer et al. (2019) [8] modified the dynamic observation error model established by Okamoto et al. [18] and applied it to all-sky assimilation using the Infrared Atmospheric Sounding Interferometer (IASI) data. Although the aforementioned *Ca* index calculation methods have been applied to some infrared satellite data assimilation systems, considering the lack of dedicated analysis on the observation errors of FY-4A/AGRI observations, it is advisable to establish observation error models tailored to FY-4A/AGRI observations. 

This study quantifies the impact of clouds on FY-4A/AGRI observations to establish observation error models suitable for the all-sky data assimilation of FY-4A/AGRI observations. There are a total of fourteen spectral bands in FY-4A/AGRI observations, but our research primarily focuses on three infrared bands, including two water vapor channels and one window channel (Band = 9, Band = 10, and Band = 14), with central wavelengths of 6.25 µm, 7.1 µm, and 13.5 µm, respectively. In addition to the two-segment linear model proposed by Okamoto et al. [18] and Harnisch et al. [20], we have developed a new three-segment linear error model that better conforms to the *O*−*B* statistics specifically for assimilating FY-4A/AGRI observations. A comparative analysis of those dynamic observation error models was conducted to identify the most suitable error model for the all-sky assimilation of FY-4A/AGRI observations.

## 2. Data and Methods

### 2.1. FY-4A/AGRI Observations

The FY-4 satellite marks a significant advancement in meteorological research and technology [21]. Compared to similar international satellites, FY-4A is equipped with 14 imaging bands in its AGRI, on par with international standards. AGRI, as the primary payload of FY-4, employs a sophisticated dual scanning mirror system to achieve precise and flexible two-dimensional pointing, enabling rapid regional scans at minute-level intervals. It utilizes an off-axis three-mirror optical system to observe Earth images frequently across more than 14 spectral bands and utilizes onboard blackbody calibration for high-frequency infrared calibration, ensuring the accuracy of observation data.

This study selected FY-4A/AGRI observations for June 2022 in southern China. The coverage area mainly included the Pan-South China region, with a latitude range from 16° N to 28° N and a longitude range from 106° E to 126° E. Southern China, located at the southernmost part of Chinese territory, belongs to a subtropical monsoon climate zone and is one of the regions most severely affected by hazardous weather phenomena in China [22]. During the summer, Southern China experiences high temperatures and humidity, with frequent occurrences of heavy rain, thunderstorms, and typhoons from May to September. Over 80% of the annual rainfall occurs during this period, significantly impacting local transportation, infrastructure, and agriculture [23]. Therefore, effectively utilizing FY-4A satellite data to improve data assimilation and numerical forecasting capabilities in Southern China is particularly important.

### 2.2. Variational Assimilation Methods and Observation Operators

This study conducts statistical analysis based on 60 datasets aggregated from two daily samples in June 2022. The forecasting model relies on the Weather Research and Forecasting Model (WRF), a mesoscale numerical weather prediction model. The assimilation system is the WRF Data Assimilation (WRFDA) three-dimensional variational assimilation system, which is compatible with the WRF model. The observation operator used for calculating background radiances is the Radiative Transfer for TOVS (RTTOV) model version 13.0 [24]. Since the publicly released version 4.0.2 of WRFDA lacks the capability to assimilate cloud-affected FY4A/AGRI radiance, this study extends the functionality of the publicly released version, 4.0.2. This extension includes the simulation of RTTOV infrared cloud radiance. Cloud scattering is based on a scaling approximation scheme, and cloud cover is based on a flux scheme [25]. The assimilation background fields are derived from the 12 h forecast fields of the WRF model. The WRF model forecast is driven by the analysis and forecast fields of the Integrated Forecasting System (IFS) of the European Centre for Medium-Range Weather Forecasts (ECMWF). After FY4A/AGRI radiance data enter the WRFDA assimilation system, they undergo preprocessing steps such as quality control and bias correction before entering the calculation of observation-minus-background differences. The configurations of the WRF model, WRFDA quality control, and bias correction schemes are detailed in Wu et al. [26].

We utilized version 3.9.1 of the convection-allowing WRF-ARW model (3 km horizontal resolution) for convection and precipitation forecasts [27]. The model configuration consisted of a single domain with 702 × 503 horizontal grids and 57 vertical layers, reaching a model peak at 10 hPa. We followed the physics option set recommended for 1–4 km grid distances in the WRF model user’s guide (https://www2.mmm.ucar.edu/wrf/users/docs/user_guide_V3.9/contents.html) (accessed on 2 December 2023), with adjustments made to surface layer and boundary layer parameter schemes. Consequently, the physical parameterizations utilized in this study included the WRF Single-Moment 6-Class Microphysics Scheme (WSM6) [28], the YSU boundary layer scheme [29], the RRTMG longwave and shortwave radiation scheme [30], the unified Noah land surface scheme [31], and the revised MM5 Monin–Obukhov surface layer scheme [32]. Cumulus parameterization was deactivated.

### 2.3. Cloud-Affected Index and Error Modeling

One method to account for the impact of clouds on the *O*−*B* difference is to introduce a predictor to quantify the cloudiness in the background field and observational data. The study adopts a *Ca* index proposed by Okamoto et al. [18], which estimates the cloud effect by simultaneously considering the observational and model-equivalent cloud effects. The calculation formula is as follows:(1)$$Ca=\frac{\left|O-B_{clr}\right|+\left|B-B_{clr}\right|}{2}$$
where *O* represents the brightness temperature observed by the FY-4A/AGRI; *B* represents the brightness temperature simulated by the RTTOV model with WRF forecasts as input; and *B_clr_* represents the simulated radiance brightness temperature without considering cloud scattering.

Harnisch et al. [20] did not use the simulated radiance brightness temperature directly without considering cloud scattering, *B_clr_*. Instead, they employed a threshold, *B_lim_*, to estimate the impact of clouds on assimilation. *B_lim_* is estimated based on the difference between brightness temperatures with and without clouds (*B*−*B_clr_*). We obtained the *B_lim_* for FY-4A/AGRI based on the method proposed by Harnisch et al. [20] (see Figure 1). In this case, the *Ca* was defined as a function of the difference between *O*, *B*, and *B_lim_*, with the calculation formula as follows: (2)$$Cx=max(0,B_{lim}-B)$$
(3)$$Cy=max(0,B_{lim}-(O-bias))$$
(4)$$Ca=\frac{Cx+Cy}{2}$$
where *Ca* is the symmetric cloud effect calculated as the average of the model-equivalent cloud effect *Cx* and the observational cloud effect *Cy* with the systematic observational bias removed.

Since *Ca* describes the average cloud effect between background and observed clouds, the point of maximum standard deviation (*SD_max_*) corresponds to the maximum mismatch between background and observed clouds, which we refer to as *Ca_max_*. When *Ca* < *Ca_max_*, the *SD* of *O*−*B* increases with *Ca*, while when *Ca* > *Ca_max_*, *SD* remains constant or decreases. In existing studies, the fitting curve of *O*−*B SD* is often modeled as a two-segment linear model. Specifically, beyond *Ca_max_*, the fitting curve becomes a horizontal line [18,33]. The two-segment linear model is as follows:(5)$$f\left(Ca\right)=\left\{\begin{array}{c} A\;*\;Ca+B,\;\;\;\;\;\;Ca<{Ca}_{max}\\ {SD}_{max},\;\;\;\;\;\;Ca\geq {Ca}_{max} \end{array}\right.$$
where *A* and *B* represent the slope and intercept, respectively, of a linear function fitted for when *Ca* < *Ca_max_*.

As a dynamic error model, $f\left(Ca\right)$ can reflect the observational error characteristics across the all-sky conditions, used to correct the background field and generate a more reasonable assimilation analysis field. However, in actual assimilation processes, this assumption may not align with the characteristics observed in real sample statistics, and the actual variation in *O*−*B SD* with *Ca* may be more complex. When assimilating FY-4A/AGRI observations, the *SD* reaches its maximum value, then begins to decline, and gradually levels off at a value of *SD_con_* (corresponding to *Ca_con_*). Based on the statistical characteristics presented by samples from FY-4A/AGRI, we have established a three-segment linear model to fit the relationship between *O*−*B SD* and Ca. The established three-segment linear model is as follows: (6)$$f\left(Ca\right)=\left\{\begin{array}{c} A*Ca+B,\;\;\;\;\;\;Ca<{Ca}_{max}\\ C*Ca+D,\;{Ca}_{max}\leq Ca<{Ca}_{con}\\ {SD}_{con},\;\;\;\;\;\;Ca\geq {Ca}_{con} \end{array}\right.$$
where *C* and *D* represent the slope and intercept, respectively, of a linear function fitted for when ${Ca}_{max}\leq Ca<{Ca}_{con}$.

For each *Ca*, we have developed two dynamic observation error models: a two-segment linear model and a three-segment linear model. In total, we have developed four types of observation error models specifically for assimilating FY-4A/AGRI observations.

## 3. Results and Discussion

### 3.1. Statistical Analysis of O−B

Figure 2 depicts the distributions of *O*−*B* for three infrared bands (Band = 9, Band = 10, and Band = 14) with central wavelengths of 6.25 µm, 7.1 µm, and 13.5 µm, respectively. While the *O*−*B* distributions for the three infrared bands exhibit similar overall characteristics, there are differences in detail. For most regions, high positive biases in *O*−*B* are rare, with more occurrences of high negative biases. This indicates that in the background field, there is a tendency for cloud top forecasts to be underestimated (i.e., cloud tops are predicted to be lower than they actually are), with the likelihood of underestimation being greater than that of overestimation. As a result, the probability distribution of *O*−*B* may exhibit asymmetrical features.

Statistical analysis reveals that the mean *O*−*B* values for the three infrared bands are −3.6 K, −5.7 K, and −9.23 K, respectively, with standard deviations of 7.8 K, 10.9 K, and 14.6 K. The mean and standard deviation of the three infrared bands increase successively, related to the severity of cloud influence on each band. The 6.25 µm water vapor band primarily detects upper tropospheric water vapor and clouds whose cloud tops are located above the upper tropospheric water vapor. The 7.1 µm water vapor band primarily detects mid-tropospheric water vapor and clouds whose cloud tops are located above the mid-tropospheric water vapor. The 13.5 µm band primarily detects near-surface temperatures and clouds at various altitudes. Therefore, the degree of cloud influence increases successively across the three bands. The greater the impact of clouds and the more complex the cloud types, the greater the difficulty in correcting the distribution of *O*−*B* to a Gaussian distribution, leading to increased assimilation challenges.

### 3.2. Cloud Effect Index and Error Modeling

To evaluate cloud impacts on the three infrared bands, this study selected two *Ca* indices, one based on the method proposed by Okamoto et al. [18] and the other based on the method by Harnisch et al. [20] (Figure 3). Based on the relationship between *Ca* and *O*−*B*, observation error models were established to correct the distribution of *O*−*B* to better fit a Gaussian distribution. Combining Figure 2 and Figure 3, it can be generally observed that when *O*−*B* exhibits high biases, both *Ca* indices tend to increase simultaneously. The horizontal distribution of the two *Ca* indices shows a high degree of consistency, effectively indicating positions with significant cloud influence, i.e., locations with large absolute values of *O*−*B* biases. Statistical analysis revealed that for the three infrared bands, the mean *Ca* values based on the method by Harnisch et al. [20] were 4.6 K, 7.3 K, and 10.5 K, with standard deviations of 5.4 K, 8.4 K, and 11.5 K, respectively. Meanwhile, the mean *Ca* values based on the method by Okamoto et al. [18] were 3.8 K, 5.8 K, and 9.8 K, with standard deviations of 4.2 K, 7.0 K, and 10.9 K, respectively. The mean and standard deviation of the two *Ca* indices for the three infrared bands increased successively, following a similar pattern to the changes in *O*−*B* for the three infrared bands. It can be concluded that clouds are an important factor influencing the distribution of *O*−*B*.

Based on the two *Ca* indices, observation error models were established for the three infrared bands. Figure 4 illustrates the relationship between *Ca* and *O*−*B* under the two methods, where the mean and variance of *O*−*B* are statistically computed within intervals of *Ca* of 1 K. It can be observed that for Band 14, the distributions of *Ca* and *O*−*B* are relatively close under both methods (Figure 4c,f), while for the other two water vapor bands, there are certain differences between the two indices. *Ca* distributions based on the method by Harnisch et al. [20] are more concentrated towards the edges. Statistical analysis revealed that for the three infrared bands, the minimum values of *O*−*B SD* based on the method by Harnisch et al. [20] were 1.7 K, 1.3 K, and 1.3 K, while the maximum values of *SD* were 11.1 K, 19.2 K, and 26.2 K, respectively. Meanwhile, for the method by Okamoto et al. [18], the minimum values of *O*−*B SD* for the three infrared bands were 1.4 K, 2.1 K, and 1.7 K, while the maximum values of *SD* were 11.6 K, 18.8 K, and 23.8 K, respectively.

From the statistical results in Figure 4, it is evident that the *SD* of *O*−*B* increases with *Ca* until it reaches a certain threshold, after which it starts to decrease before eventually leveling off. Therefore, the two-segment linear model fails to accurately reflect the statistical characteristics observed in the actual samples. Additionally, since *Ca* is based on a spectral cloud-affected index, it may not fully represent the actual magnitude of cloud influence. Consequently, the actual variation in *O*−*B SD* becomes complex as *Ca* increases. Building upon these observations, the study establishes a three-segment linear model to fit the relationship between *O−B SD* and *Ca* based on the statistical characteristics observed in the actual samples (Figure 4).

### 3.3. Statistical Analysis of O−B

Based on the two *Ca* indices, two two-segment linear observation error models were established, correcting the probability density distribution of *O*−*B* in FY-4A/AGRI observations assimilation, as shown in Figure 5. The red line represents the original probability density distribution of *O*−*B*, the blue line represents the probability density distribution of *O*−*B* corrected based on the observation error model, and the dashed line represents the standard Gaussian distribution. It can be observed that the correction effect of the two-segment linear observation error model established based on Okamoto et al. [18]’s method is not very satisfactory (Figure 5d–f). In this case, dynamic error estimation leads to *O*−*B* deviating from the Gaussian distribution and exhibiting sharp edges. Some other two-segment linear observation error models established using Okamoto et al.’s method [18] have encountered similar issues, leading to the introduction of a minimum error estimation for data with smaller *Ca* mean values, but still prone to bimodal distributions in *O*−*B* [20]. The correction of the probability density distribution by the two-segment linear observation error model established based on Harnisch et al.’s method [20] does not exhibit obvious bimodal distributions, and the distribution curves are relatively smooth. The two two-segment linear observation error models established based on the two *Ca* indices do not strongly adhere to Gaussianity in the corrected probability density distribution of *O*−*B*. Particularly for Band 14 (Figure 5c,f), the distribution notably deviates from the Gaussian distribution.

Subsequently, two three-segment linear observation error models were established based on the two *Ca* indices, correcting the probability density distribution of *O*−*B* in FY-4A/AGRI observation assimilation, as shown in Figure 6. Compared to the two-segment linear observation error model based on Okamoto et al. [18]’s method (Figure 5d–f), although there is still a bimodal distribution structure, the Gaussianity of the probability density distribution of the three-segment linear observation error model (Figure 6d–f) has been significantly improved. Compared to the two-segment linear observation error model based on Harnisch et al.’s method [20] (Figure 5a–c), the Gaussianity of the probability density distribution of the three-segment linear observation error model has also been comparatively improved. The three-segment linear observation error model fits the actual relationship between *SD* and *Ca* better, resulting in a more effective observation error model.

## 4. Conclusions

This study analyzed the observation errors of the FY-4A/AGRI infrared channels during all-sky data assimilation, focusing on three infrared bands: Band 9 (center wavelength 6.25 µm), Band 10 (center wavelength 7.1 µm), and Band 14 (center wavelength 13.5 µm). Four dynamic observation error models were developed to provide more accurate error estimates for assimilating the brightness temperature of the FY-4A/AGRI infrared channels. Experimental results showed that the observation errors of these three bands exhibited similar distribution characteristics, mainly characterized by high negative biases. Specifically, the mean *O*−*B* values were −3.6 K, −5.7 K, and −9.23 K, respectively. When correcting the *O*−*B* probability density distribution of FY-4A/AGRI observation assimilation based on two different *Ca*, the two-segment linear model based on the Okamoto et al. method [18] exhibited sharp edges and bimodal distributions, while the Gaussianity of the *O*−*B* probability density distribution corrected by the two-segment linear model based on the Harnisch et al. method [20] was not strong, especially for Band 14. Subsequently, the three-segment linear observation error model made progress in correcting the *O*−*B* probability density distribution of FY-4A/AGRI observation assimilation. Compared to the two-segment linear model, the Gaussianity of the probability density distribution of the three-segment linear model was significantly improved. This indicates that the three-segment linear model is closer to the actual relationship between *SD* and *Ca*, thereby more effectively correcting observation errors and improving the accuracy of data assimilation. This finding is of great significance for improving the assimilation of FY-4A/AGRI observations under all-sky conditions, contributing to the accuracy and reliability of weather forecasting.

## Figures and Tables

**Figure 1 sensors-24-02572-f001:**
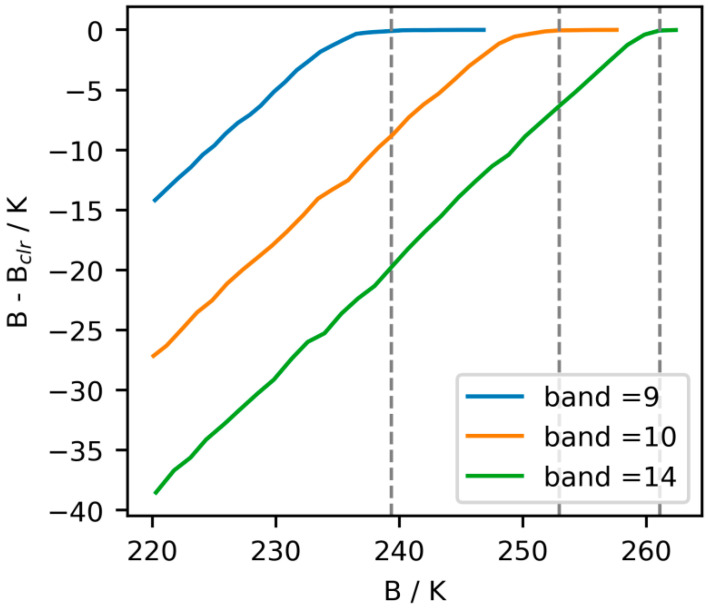
Relationship of the difference between the brightness temperatures with and without cloud scattering (*B*−*B_clr_*) and background brightness (*B*). The vertical dashed line indicates the threshold *B_lim_*.

**Figure 2 sensors-24-02572-f002:**
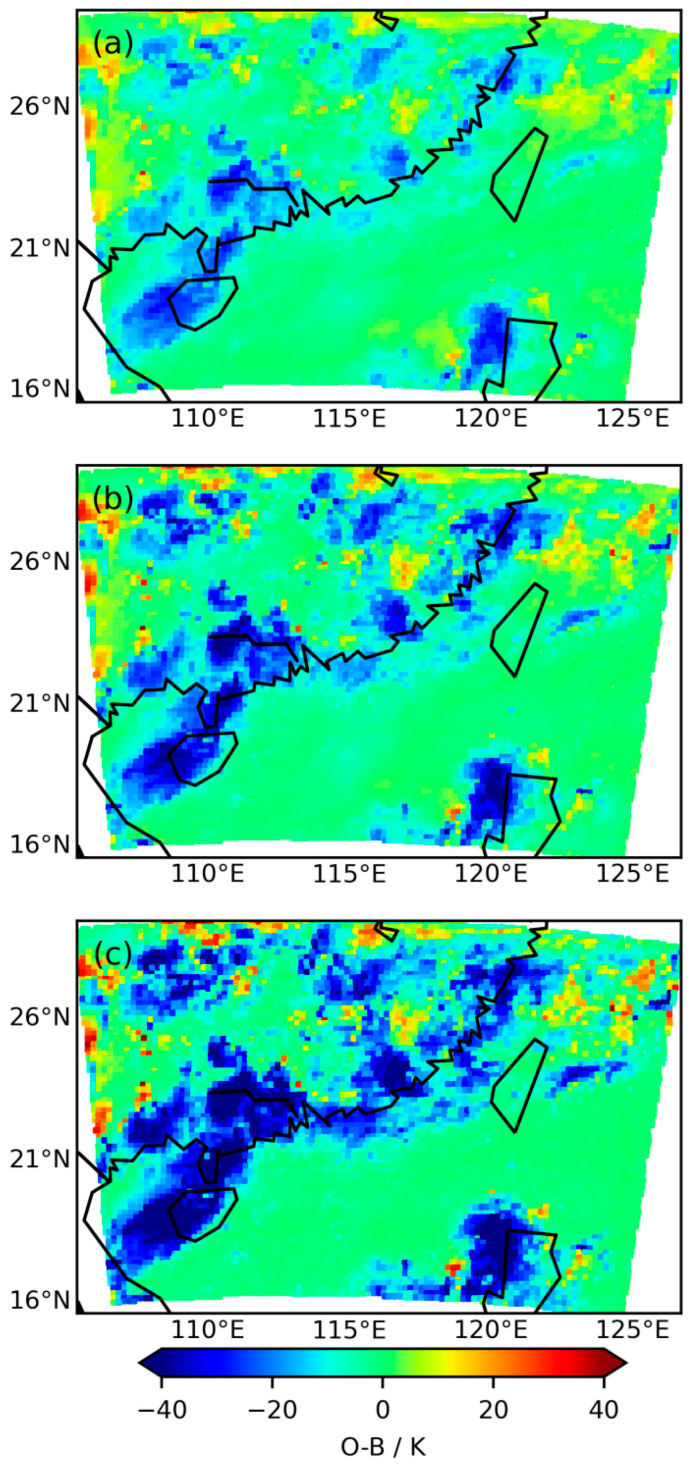
The observation-minus-background (*O*−*B*) distributions for three infrared water vapor bands on 30 June 2022 at 00:00 UTC. (**a**) Band = 9, (**b**) Band = 10, and (**c**) Band = 14. Their center wavelengths are 6.25 µm, 7.1 µm, and 13.5 µm, respectively.

**Figure 3 sensors-24-02572-f003:**
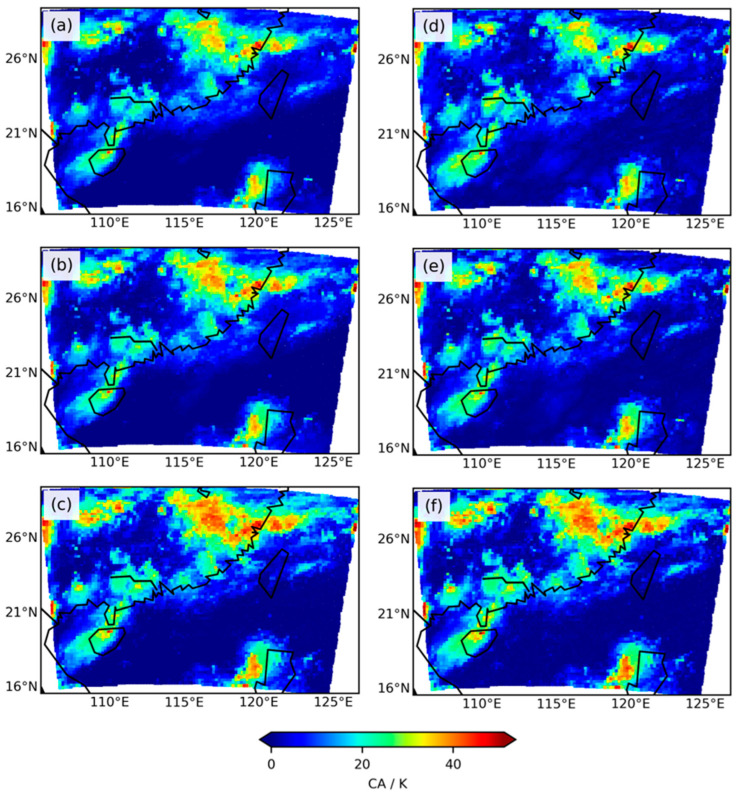
Cloud impacts in the three infrared water vapor bands are assessed based on two cloud-affected indices (*Ca*). Rows 1 to 3 denote bands 9, 10, and 14. Panel (**a**–**c**) show the error models obtained using the Okamoto et al. method [18]. Panel (**d**–**f**) show the error models obtained using the Harnisch et al. method [20].

**Figure 4 sensors-24-02572-f004:**
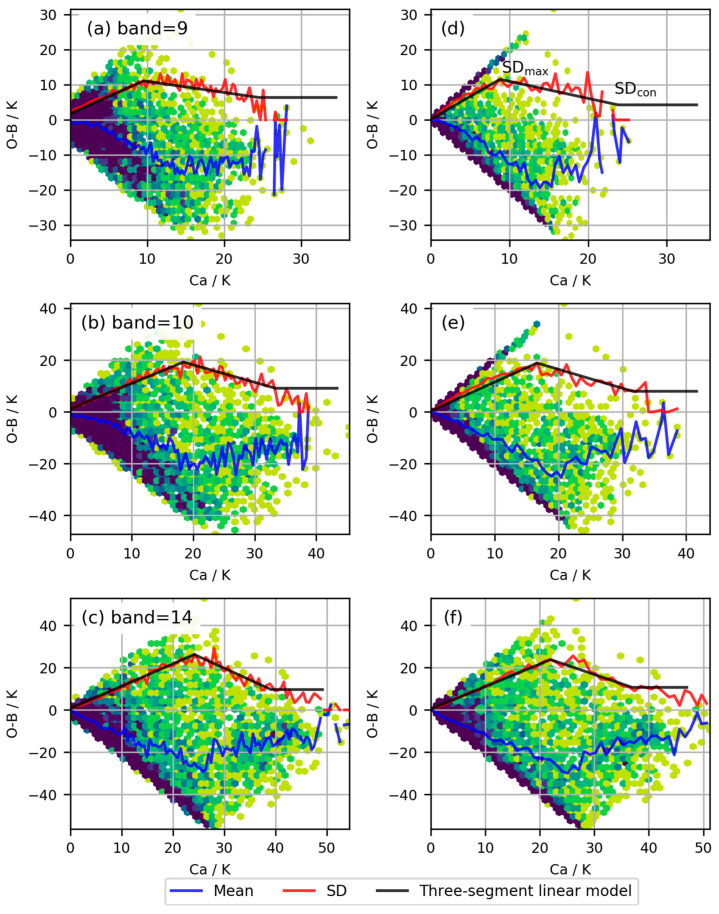
Observation errors modelled for the three infrared water vapor bands based on two types of *Ca*. Rows 1 to 3 denote bands 9, 10, and 14. Panel (**a**–**c**) show the error models obtained using the Okamoto et al. method [18]. Panel (**d**–**f**) show the error models obtained using the Harnisch et al. method [20]. When assimilating FY-4A/AGRI observations, the *SD* reaches its maximum value (*SD_max_*), then begins to decline, and gradually levels off at a value of *SD_con_*.

**Figure 5 sensors-24-02572-f005:**
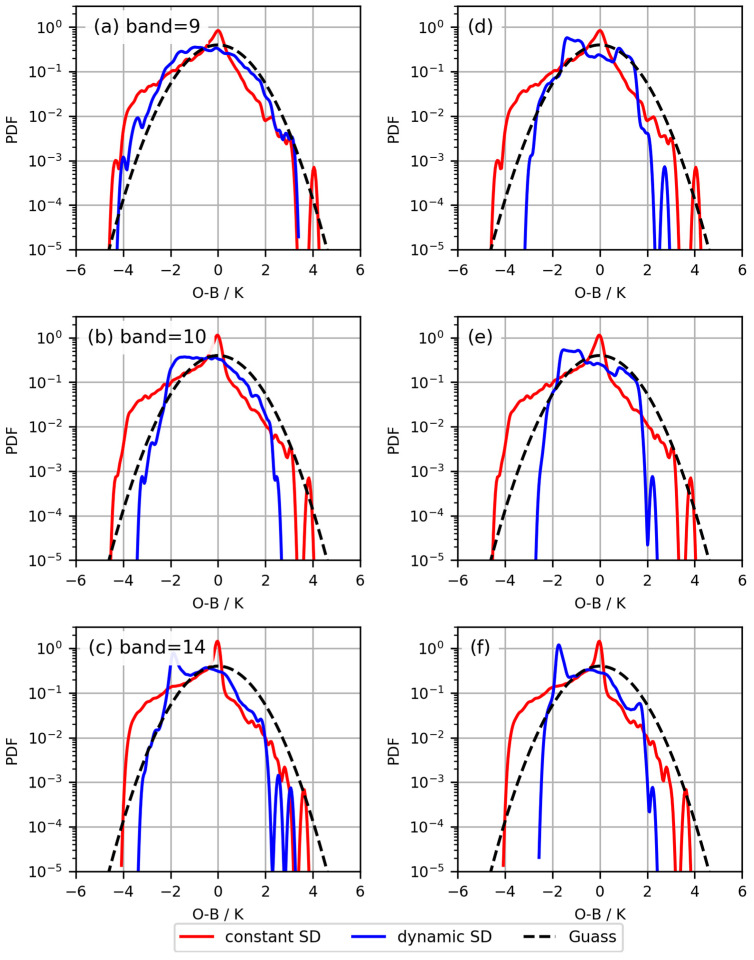
The two-segment linear observation error models based on the two *Ca*, corrected for the *O*−*B* probability density distribution (PDF). Rows 1 to 3 denote bands 9, 10, and 14. Panel (**a**–**c**) show the error models obtained using the Okamoto et al. method [18]. Panel (**d**–**f**) show the error models obtained using the Harnisch et al. method [20].

**Figure 6 sensors-24-02572-f006:**
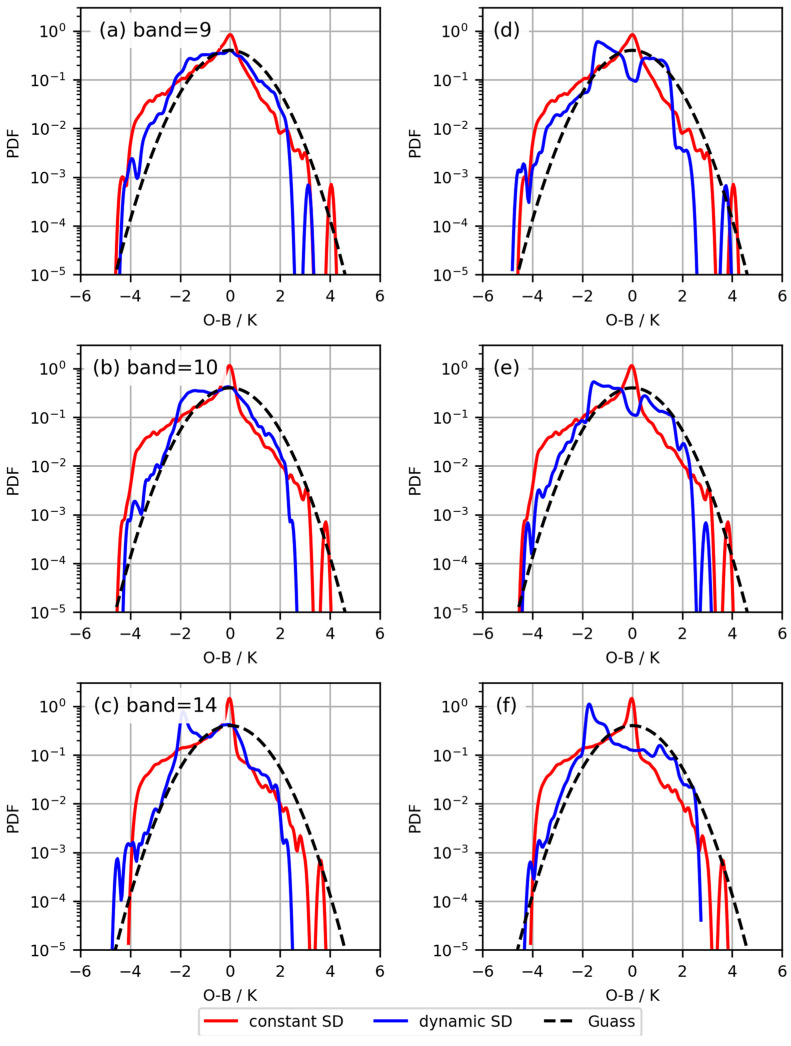
The three-segment linear observation error models based on the two *Ca*, corrected for the *O*−*B* probability density distribution. Rows 1 to 3 denote bands 9, 10, and 14. Panel (**a**–**c**) show the error models obtained using the Okamoto et al. method [18]. Panel (**d**–**f**) show the error models obtained using the Harnisch et al. method [20].

## Data Availability

The ECMWF IFS analysis and forecast data were provided by the Shenzhen Meteorological Bureau of China Meteorological Administration. The FY-4A AGRI radiance data are available from the China National Satellite Meteorological Centre by registration (https://satellite.nsmc.org.cn/PortalSite/) (accessed on 6 December 2023). The WRF and WRFDA models were developed by American National Center for Atmospheric Research (NCAR) and can be downloaded from https://github.com/wrf-model/WRF (accessed on 15 December 2023).

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
