# Peer review of "Error Model for the Assimilation of All-Sky FY-4A/AGRI Infrared Radiance Observations"

_sensors, 2024, doi:10.3390/s24082572_

Round 1
Reviewer 1 Report
Comments and Suggestions for Authors
Dear colleagues!
Thank you for the opportunity to review this article, which presents the results of studies of the observation error of the FY-4A/AGRI infrared channels when processing data from the entire sky with an emphasis on infrared channels. The authors have developed four models of dynamic observation errors, according to the results of processing which observation errors using a three-segment linear model showed the best results. This research is very important for improving the accuracy and reliability of weather forecasting. The article is written in an accessible manner and the results presented in it certainly deserve publication in this journal. The article is good presented and has a clear structure which simplifies the understanding of the research essence,but I still want to suggest mentioning the following:
The Figure 4. Observation errors modelled for the three infrared water vapour bands based on two types of Ca on (a),(b),(c) the x-axis signatures contain both capital letters.
Figure 4 -6 (Rows 1 to 3 denote bands 9, 10, and 14, the first and second columns show the Ca obtained ..) What rows, bands and columns are we talking about? On The figures show statistical curves and a set of data array points. If you mean a row containing figures a and d, and under the columns you mean a set of figures a, b, c, then it is better to indicate the letter of a particular figure in the caption and explanation.
Best regards
Reviewer 2 Report
Comments and Suggestions for Authors
General comments:
The authors conducted a study and discussion on the observation error model used in the data assimilation of the three infrared channel radiance data of FY4A/AGRI under all-sky conditions, proposing a three-segment linear model, which is of significant importance. However, there is still room for improvement in the introduction and description of the methodology. The logical coherence, readability, and completeness of the method description in the paper need to be enhanced. Only by clearly describing the methodology can I better assess the rationality and innovation of the paper's methods and results.
Specific comments:
The narrative logic from line 55 to line 77 needs to be reorganized. It is necessary to introduce the concept and role of observation error models more reasonably, as well as their development history.
Other relevant information about the WRF model, such as the grid, additional parameterization schemes, etc., also needs to be provided.
For the description of the observation error model:
Line 162, the explanation for Cy and bias is not clear enough, which is very confusing.
Line 163, the influence of clouds on brightness temperature is not only determined by cloud coverage but also by other parameters such as cloud altitude, which have significant effects. Therefore, the statement ‘Ca describes the average cloud amount between’ is incorrect.
Line 164, what is ‘the point of maximum standard deviation’? It should be described.
The linear model presents a way to calculate f(Ca), but what is f(Ca)? What is the utility of this variable? Where should it be used to implement data assimilation? The paper completely fails to explain.
Another important issue is that the author uses model clear-sky results as the background field. It is necessary to provide a comparison between the model output of radiation/brightness temperature under clear-sky conditions and satellite observations to validate the reliability of using model clear-sky results as the background field.
Minor comments / typos:
Line 13, a comma is missing between the words ‘uncertainty’ and ‘and’. Without the comma, the sentence is confusing.
Line 29, after mentioning the ‘Advanced Geostationary Radiation Imager’, it should be noted that the abbreviation is ‘AGRI’.
Lines 91 to 100, payloads that are unrelated to the research topic of the paper do not need to be introduced.
The distance between the position of Figure 1 and its corresponding description in the main text is somewhat distant. Besides, it is necessary to introduce how the data in Figure 1 was obtained.
Line 145, the reference note is missing after ‘Okamoto et al.’. Same for line 153.
Line 157, ‘O-B’ or ‘O/B’ should be unified.
Round 2
Reviewer 2 Report
Comments and Suggestions for Authors
The authors have addressed all my comments, and I recommend publication of the manuscript.